# Fluorescent Cytochemical Detection of Polyphosphates Associated with Human Platelets

**DOI:** 10.3390/ijms22031040

**Published:** 2021-01-21

**Authors:** Atsushi Sato, Hachidai Aizawa, Tetsuhiro Tsujino, Kazushige Isobe, Taisuke Watanabe, Yutaka Kitamura, Tomoyuki Kawase

**Affiliations:** 1Collaborative Research Group, Tokyo Plastic Dental Society, Tokyo 114-0002, Japan; atu.net@nifty.com (A.S.); sarusaru@mx6.mesh.ne.jp (H.A.); tetsudds@gmail.com (T.T.); kaz-iso@tc4.so-net.ne.jp (K.I.); 2Division of Anatomy and Cell Biology of the Hard Tissue, Institute of Medicine and Dentistry, Niigata University, Niigata 951-8514, Japan; watatai@mui.biglobe.ne.jp; 3Matsumoto Dental University Hospital, Shiojiri 399-0781, Japan; shinshu-osic@mbn.nifty.com; 4Division of Oral Bioengineering, Institute of Medicine and Dentistry, Niigata University, Niigata 951-8514, Japan

**Keywords:** polyphosphates, platelets, calcium activation, 4′,6-diamidino-2-phenylindole, alkaline phosphatase

## Abstract

Polyphosphate (polyP) is released from activated platelets and activates the intrinsic coagulation pathway. However, polyP may also be involved in various pathophysiological functions related to platelets. To clarify these functions, we established a cytochemical method to reproducibly visualize polyP in platelets. Platelets obtained from healthy non-smoking donors were suspended in phosphate-buffered saline and quickly immobilized on glass slides using a Cytospin. After fixation and membrane permeabilization, platelets were treated with 4′,6- diamidino-2-phenylindole (DAPI) and examined using a fluorescence microscope with a blue-violet excitation filter block (BV-2A). Fixed platelets were also subjected to immunocytochemical examination to visualize serotonin distribution. Under the optimized conditions for polyP visualization, immobilized platelets were fixed with 10% neutral-buffered formalin for 4 h or longer and treated with DAPI at a concentration of 10 µg/mL in 0.02% saponin- or 0.1% Tween-20-containing Hanks balanced salt solution as a permeabilization buffer for 30 min at room temperature (22–25 °C). Based on the results obtained by using activated platelets, treatment with alkaline phosphatases, and serotonin release, the DAPI^+^ targets were identified as polyP. Therefore, this cytochemical method is useful for determining the amount and distribution of polyP in platelets.

## 1. Introduction

Regenerative therapy using platelet-rich plasma (PRP) is inspired by the process of wound healing [1,2,3]. This therapy appears to be primarily dependent on biomolecules, including growth factors, released from platelets [4]. Specifically, these biomolecules are released upon activation and are immediately entrapped by newly formed fibrin fibers surrounding the platelets and do not simply diffuse away. This process forms an efficient controlled release system that retains the platelet-derived biomolecules at higher levels, thereby continuously acting on the surrounding cells involved in tissue regeneration.

The fibrin meshwork is constructed mainly, but not solely, by coagulation factors contained in the coagulation pathway. In the process of clot formation in peripheral tissues, or in the absence of plasma fibrinogen [5], platelet aggregation precedes fibrin formation. Therefore, platelets activated by external stimuli release biomolecules and expose phosphatidylserine on their surface [6] to induce aggregation as well as coagulation. Simultaneously, newly formed fibrin fibers activate platelets to further grow a blood clot via a positive feedback loop. Taken together, these results indicate that the function of platelets in PRP therapy includes, but is not limited to, their supply of growth factors. Platelets take the initiative to control their growth factor retention and release.

Polyphosphate (polyP) is a biomolecule stored in dense granules of platelets. This anionic polymer in platelets comprises three to 1000 monomers of phosphate and is involved in several pathophysiological functions, including coagulation and biomineralization [7,8,9,10], although a degradation product of polyP, pyrophosphate, possibly acts as a key inhibitor of biomineralization [11]. In terms of coagulation, polyP activates factor XII and triggers the intrinsic coagulation pathway [12]. Therefore, it is generally accepted that platelets release polyP along with several other coagulation factors upon activation to induce clot formation. However, due to technical limitations of polyP quantification or visualization, it is currently poorly understood how polyP is metabolized and mobilized in platelets.

To fully elucidate the role of polyP in platelet functions, it is necessary to accurately determine polyP concentrations and cytologically detect its distribution. In earlier studies [13], basic dyes such as toluidine blue, methylene blue, neutral red, and tetracycline were used to visualize polyP; however, their lower specificity, sensitivity, and reproducibility are major shortcomings. An alternative method using the polyP-binding domain of *E. coli* exopolyphosphatase fused with an Xpress tag has higher specificity and sensitivity but is not easily available commercially [13]. Efforts have been made to develop specific antibodies against polyP; however, because of their simple repeated framework, reliable antibodies are not currently available. Instead, a probe with fairly good sensitivity and specificity, 4′,6-diamidino-2-phenylindole (DAPI), has been used to detect polyP in biochemical and cytological methods, mainly in the field of microbiology [13]. Many further efforts have been made to develop specific antibodies against polyP; however, owing to their simple repeated framework, reliable antibodies are not currently available. Instead, a probe with a fairly good sensitivity and specificity, 4′,6-diamidino-2-phenylindole (DAPI), has long been used to detect polyP in biochemical and cytological methods, mainly in the field of microbiology [13]. However, DAPI has rarely been utilized in the study of polyP in mammalian cells [14]. To develop and establish a cytochemical method to visualize polyP distribution in platelets, we optimized the experimental conditions for DAPI staining.

## 2. Results

Figure 1 shows the effects of fixation time on polyP detection in platelets. Platelets suspended in PBS were immobilized on glass slides and fixed in 10% neutral-buffered formalin for 1, 2, 4, or 18 h. In general, fixation time with formalin should be minimized for immunofluorescence staining to avoid unexpected epitope masking. When examining serotonin distribution, platelets were fixed for 1 h. However, for polyP, such a short fixation appeared insufficient to retain polyP on the specimen. These results clarified that at least 2 h of fixation was needed to reproducibly detect polyP and that prolonging the fixation time gradually increased the levels of polyP that were detected. A parametric test revealed that a significant difference was obtained between the 1 h and the 18 h group.

Figure 2 shows the effects of detergent type on polyP detection in platelets. Although DAPI is a membrane-permeable dye, fixation with formalin blocks its transmembrane migration. In addition, more vivid phalloidin staining requires membrane permeabilization. Therefore, it is important to select appropriate detergents. We compared the performances of three popular detergents and found that Tween-20 and saponin were appropriate at individualized concentrations. A parametric test revealed that Tween-20 and saponin provided significantly higher values than Triton X-100 and no addition of detergent in HBSS.

Figure 3 shows the effects of buffer type on polyP detection in platelets. The major differences between PBS and HBSS were the presence of d-glucose and sodium hydrogen carbonate in the latter. Although possible causal relationships and statistically significant differences were not observed in any of the comparisons, HBSS seemed to support clearer polyP detection with both Tween-20 and saponin. The HBSS groups provided higher values than the PBS groups; however, significant differences were not observed either by a parametric or a non-parametric test.

Figure 4 and Figure 5 show the effects of DAPI concentrations on polyP detection in platelets. Compared to DNA staining, PolyP staining may require higher concentrations of DAPI. However, to minimize unidentified non-specific binding and reduce the expense of unnecessary dye without sacrificing data reliability, we optimized the concentration of DAPI. DAPI at 2.5, 5.0, 10, and 20 μg/mL (green) was examined in both 0.1% Tween-20-containing HBSS (Tw-HBSS) (Figure 4) and 0.02% saponin-containing HBSS (Spn-HBSS) (Figure 5). A parametric test revealed that in Figure 4, DAPI dose-dependently increased the values and that the rate of increase was suppressed between 10 and 20 μg/mL, and that in Figure 5, the values reached the plateau at 10 μg/mL DAPI. Thus, maximal staining was commonly observed at 20 and 10 μg/mL DAPI in Tween-20 and saponin, respectively. Therefore, to reduce possible non-specific binding, we used 10 μg/mL DAPI in all the subsequent experiments.

Figure 6 and Figure 7 show the effects of CaCl_2_ on the detection of polyP in platelets. Because platelets suspended in Ca^2+^-free PBS are activated directly by adding exogenous CaCl_2_ suspended in PBS [5,13], polyP mobilization was examined in activated platelets. In the control Tw-HBSS, polyP was observed mainly as small particles and colocalized in platelets (Figure 6). Activation reduced such particles associated with platelets and instead increased them in the extra-platelet spaces, finally diffusing widely without granular shapes. In Spn-HBSS, a similar tendency was observed; however, the diffusion of particles occurred somewhat faster than in Tw-HBSS (Figure 7).

Figure 8 shows the effects of the fixative types on polyP detection in platelets. Formalin (10%) had a much higher fixation ability, and there was weak membrane permeabilization caused by methanol contamination. In contrast, ThromboFix fixes platelets without membrane permeabilization or structural destruction. In HBSS, in the absence of any detergents, granular polyP was observed in platelets in the control, while diffused polyP was observed extracellularly in platelets activated with 0.1% CaCl_2_ for 60 min. In ThromboFix, granular polyP was not observed in the control group. However, diffused polyP was widely visualized in extra-platelet spaces.

Figure 9 shows the effects of ALP on the detection of polyP in platelets. After 4 h of fixation with 10% neutral-buffered formalin, control platelets were treated with 0.04 U/μL ALP in Mg^2+^-containing alkaline buffer for 24 h and then stained with DAPI in Tw-HBSS. As expected, ALP treatment almost completely eliminated granular and diffused polyP. Both parametric and non-parametric tests revealed that ALP treatment significantly reduced the value.

Figure 10 shows the effects of CaCl2 on the distribution of polyP and serotonin in platelets. Under the conditions used for polyP detection, platelets were activated by 0.1% CaCl_2_ and subjected to immunofluorescence examination of serotonin, which is stored in dense granules of platelets along with polyP [15]. In control platelets, serotonin was as visible as the granular substances, similar to polyP in platelets, and diffused in the extra-platelet spaces when activated by 0.1% CaCl_2_ at 60 min.

Figure 11 shows the effects of ADP on the distribution of polyP and serotonin in platelets. ADP, an activator of platelets [16], was used to examine the similarity in the mobilization of polyP and serotonin. In the control, as observed in Figure 10, granular polyP and serotonin were observed in platelets. When activated by 5 mM ADP for 60 min, the particles disappeared, but the typical diffusion observed above was not detected.

## 3. Discussion

Cytochemical visualization of polyP with DAPI has been reported previously [13,14,17,18]. However, to our knowledge, its validity has not yet been fully investigated in platelets. The main purpose of this study was to determine the specificity of DAPI staining in platelet suspensions. For this purpose, we established platelet models that underwent time-course activation using exogenously added CaCl_2_ and treatment with ALP. As expected, DAPI^+^ targets that we detected in Ca^2+^-activated platelets decreased with time of incubation, and the extra-platelet spaces were widely and faintly stained with DAPI. In addition, ALP treatment almost completely eliminated the DAPI^+^ targets. Therefore, although the possibility of artifacts, such as non-specific binding or specific binding to similar compounds, was not completely excluded, we concluded that the DAPI^+^ targets were polyP, and that the specificity was sufficiently high to further optimize the experimental conditions. This included the choice of fixatives, fixation time, method of membrane permeabilization, and choice of staining buffer.

For fixation, our results indicated that 10% formalin was most suitable for stabilizing platelets and that polyP was better retained on the specimen, as the fixation time was prolonged. Therefore, the platelets were fixed in 10% formalin for at least 4 h at room temperature. For membrane permeabilization, our results indicated that Triton X-100 at the concentration used (0.1%) may be too strong for surfactant action to retain some important compounds involved in DAPI visualization. When treatment was limited to 30 min, saponin (0.02%) was slightly more effective than Tween-20 (0.1%) in HBSS. PBS was not acceptable because of its low reproducibility.

We believe that these cytochemical findings provide enough “semi-quantitative” data to examine the mobilization of polyP in the activation process. For more accurate quantification, it may be better to normalize the DAPI values quantified through image analysis using other constant values, such as those of housekeeping genes in electrophoresis. Since we could not find conventionally used markers, we attempted to normalize DAPI^+^ targets using phalloidin-stained F-actin as a possible candidate marker. However, as described previously [19,20], we confirmed its strong dependency on the activation status of platelets and its inappropriateness as a marker for normalization. Instead, the quantification of DAPI^+^ targets in platelets using the fluorometric method is currently under investigation.

To confirm the specificity of DAPI^+^ targets via the indirect method, we performed an additional experiment on serotonin distribution. Because serotonin is stored along with polyP in dense platelet granules, platelets may release both biomolecules upon activation. PolyP was released synchronously with serotonin in 0.1% CaCl_2_. Similar findings were obtained when the platelets were stimulated with 5 mM ADP. Therefore, we concluded with high probability that the DAPI^+^ targets observed by this cytochemical method were polyP in anuclear cells and platelets.

## 4. Clinical Relevance 

Many clinicians performing platelet-rich fibrin (PRF) therapy may have experienced non-coagulated blood samples during PRF preparation. There are several possible explanations for this phenomenon. A shortage of polyP is one such possibility; however, there are few studies available that support this mechanism. The current cytochemical method is not highly quantitative but can semi-quantitatively determine the polyP content in platelets without expensive equipment or special reagents. Therefore, in some cases, it will be possible to identify the involvement of polyP in tissue regeneration in the near future.

## 5. Materials and Methods 

### 5.1. Preparation of Pure PRP

Blood samples were collected from six non-smoking, healthy, male volunteers aged 46–62 years. Despite having lifestyle-related diseases and taking medications, these donors (i.e., our team members and relatives) had no limitations on their daily living activities. These donors were also free from HIV, HBV, HCV, or syphilis infections. A prothrombin test was performed on all blood samples using CoaguChek^®^ XS (Roche, Basel, Switzerland), and all samples were normal. Platelet and other blood cell counts were measured using a pocH 100iV automated hematology analyzer (Sysmex, Kobe, Japan).

First, ~9 mL of peripheral blood was collected in plain glass vacuum blood collection tubes (Vacutainer^®^; BD Biosciences, Franklin Lakes, NJ, USA) containing 1.5 mL of an acid–citrate–dextrose solution. [21,22]. Whole-blood samples were stored in a rotating agitator at ambient temperature and were used within 24 h. Thereafter, the samples were centrifuged using a horizontal centrifuge (Kubota, Tokyo, Japan) at 402× *g* for 8 min (soft spin). The upper plasma fraction, ~2 mm above the interface between the plasma and red blood cell fractions, was then transferred into 2 mL sample tubes and centrifuged once again using an angle-type centrifuge (Sigma Laborzentrifugen, Osterode am Harz, Germany) at 1065× *g* for 3 min (hard spin) to collect the resting platelet pellets. Afterward, the platelets were resuspended in phosphate-buffered saline (PBS) to adjust the platelet concentration to 3.0 × 10^5^/µL.

The study design and consent forms for all procedures (project identification code: 2019-0423) were approved by the Ethics Committee for Human Participants at the Niigata University School of Medicine (Niigata, Japan) and complied with the Helsinki Declaration of 1964, as revised in 2013.

### 5.2. Cytochemical Staining of PolyPs

PolyPs stored in, and released from, platelets were visualized using 4′,6-diamidino-2- phenylindole (DAPI; Dojindo, Kumamoto, Japan) [23]. In brief, platelets were immobilized on glass slides using a Cytospin 4 cytocentrifuge (Thermo Fisher Scientific, Waltham, MA, USA) and fixed with 10% neutral-buffered formalin for 1, 2, 4, or 18 h, or ThromboFix (Beckman-Coulter, Brea, CA, USA) overnight. After washing with PBS, the platelets were stained with six different DAPI-containing buffers (10 μg/mL) in the presence of Phalloidin-iFluor 555 (1:200 dilution) (Abcam, Cambridge, UK) for 30 min. PBS and Hanks balanced salt solution (HBSS) were mixed with 0.1% Tween-20 (FUJIFILM Wako Pure Chemical, Osaka, Japan), 0.02% saponin, (Sigma-Aldrich, St. Louis, MO, USA), or 0.1% Triton X-100 (FUJIFILM Wako Pure Chemical) to create these six buffers. The specimens were then washed and mounted using an antifade mounting medium (Vectashield; Vector Laboratories, Burlingame, CA, USA) and subjected to microscopic examination using a fluorescence microscope (Eclipse 80i; Nikon, Tokyo, Japan) equipped with a BV-2A filter cube (excitation filter: 400–440 nm; dichroic mirror: 455 nm; barrier filter: 470 nm) to detect DAPI and phalloidin, respectively.

### 5.3. Alkaline Phosphatase Treatment

The fixed platelets were treated with 0.04 U/μL ALP (TaKaRa Bio, Kusatsu, Japan) in an attached ALP-specific buffer at 37 °C for 24 h. The specimens were washed and stained with DAPI and phalloidin, as described above.

### 5.4. Immunocytochemical Fluorescence Staining of Serotonin

Platelets, fixed on glass slides, were blocked with BlockAce (Sumitomo-Dainippon Pharma, Osaka, Japan) in 0.1% Tween-20-containing PBS and treated with a mouse monoclonal anti-serotonin antibody (1:200 dilution) (GeneTex, Hsinchu City, Taiwan) overnight at 4 °C, followed by probing with Alexa Fluor 488-conjugated goat anti-mouse IgG H&L (Abcam) for 40 min at room temperature (22–25 °C). The specimens were mounted using Vectashield (Vector Laboratories), and serotonin was visualized using a fluorescence microscope connected to a cooled CCD camera [5].

### 5.5. Image Analysis

We randomly selected original, non-touched photomicrographs from representative single experiments (*n* = 3) and analyzed the images using the WinROOF software (version 6.0.0.; Mitani Co., Fukui, Japan). In brief, the merged images were separated into three independent RGB images. Red-channel images were directly counted using the manual counting mode of the software. After thresholds were set at inflection points that were identified visually in gamma profiles (full grayscale), the brightness of each blue-channel image was determined in the same area (109,060 pixels) of the region of interest using WinROOF. These brightness values were normalized to the platelet counts. However, this analysis was ineffective for the images of non-aggregated platelets. Therefore, we could not analyze the CaCl_2_-treated platelet populations.

### 5.6. Statistical Analysis

Data are expressed as the mean ± standard deviation. SigmaPlot (SigmaPlot 13.0; Systat Software, Inc., San Jose, CA, USA) evaluated the image analysis data as parametric by both normality and equal variance testing and suggested a one-way ANOVA followed by Bonferroni’s multiple comparisons test as the appropriate analysis. In comparison of two groups (Figure 9), Student’s t test was performed. The results of this analysis showed significant differences in some comparisons, as indicated in the figures. Differences with *p* < 0.05 were considered statistically significant.

However, as these data were derived from a limited number of samples (*n* = 3), a non-parametric analysis was also performed. We compared the median values using Kruskal–Wallis one-way analysis of variance, followed by the Steel–Dwass multiple comparison test (BellCurve for Excel; Social Survey Research Information Co., Ltd., Tokyo, Japan). No statistical differences were observed in any of the comparisons. In comparison of two groups (Figure 9), the Mann–Whitney U test was performed, and a significant difference was observed.

## 6. Conclusions

This cytochemical visualization method enables the semi-quantitative determination of intra-platelet polyP levels. This technical development will be useful for evaluating the suitability of blood samples for PRF preparation and studying polyP mobilization in the process of PRF preparation.

## Figures and Tables

**Figure 1 ijms-22-01040-f001:**
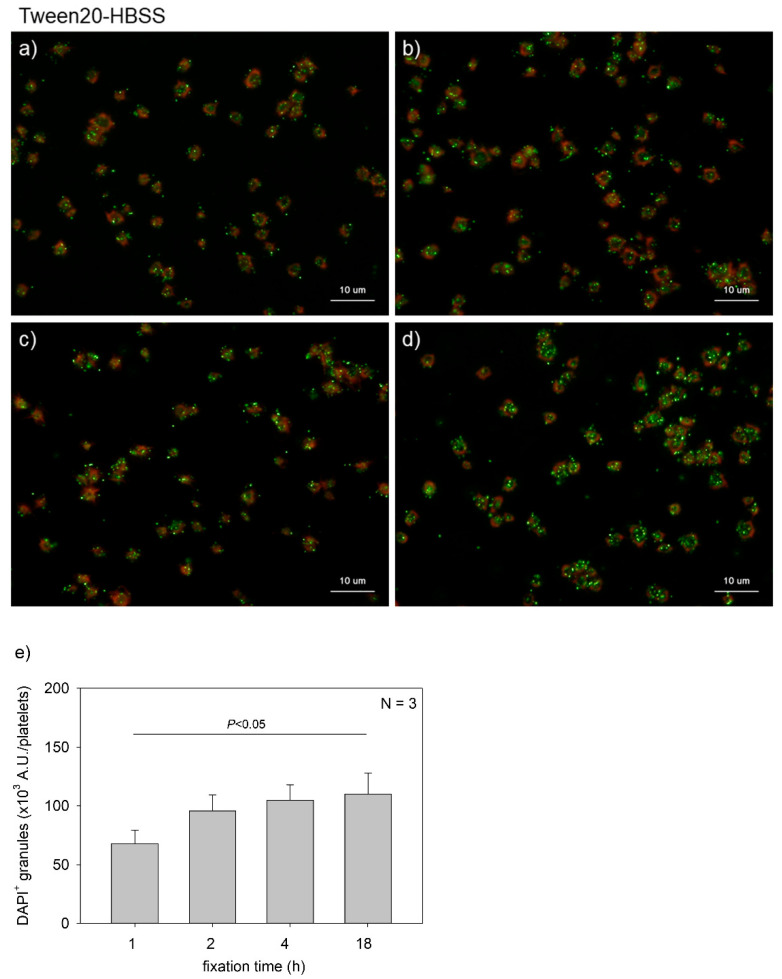
Effects of fixation time on polyphosphate detection in platelets. Platelets suspended in phosphate-buffered saline were immobilized on glass slides using a Cytospin and fixed in 10% neutral-buffered formalin for (**a**) 1 h, (**b**) 2 h, (**c**) 4 h, or (**d**) 18 h. Platelets were then treated with 4′,6-diamidino-2-phenylindole (DAPI; green) and phalloidin (red) in 0.1% Tween-20-containing Hanks balanced salt solution and examined using a fluorescence microscope. (**e**) Quantification of DAPI staining in the fluorescence images (above). A significant difference was obtained via a parametric test but not by using a non-parametric test.

**Figure 2 ijms-22-01040-f002:**
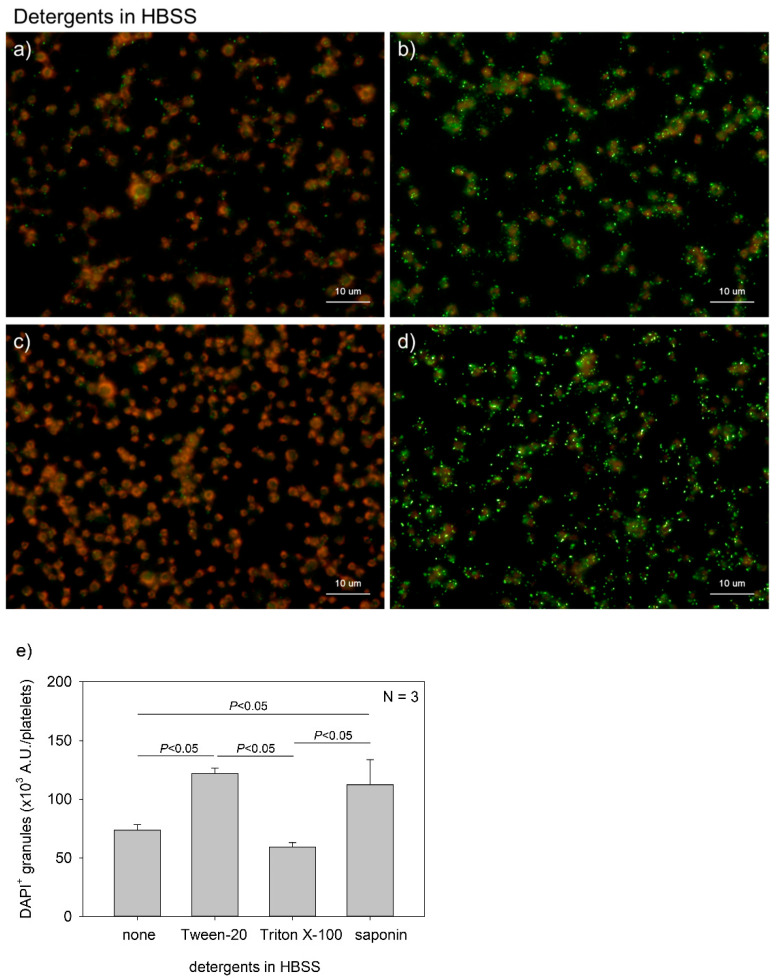
Effects of detergent types on polyphosphate detection in platelets. Platelets immobilized on glass slides using a Cytospin were fixed in 10% neutral-buffered formalin for 4 h, then treated with 4′,6-diamidino-2-phenylindole (green) and phalloidin (red) in (**a**) detergent-free Hanks balanced salt solution (HBSS), (**b**) 0.1% Tween-20-containing HBSS, (**c**) 0.1% Triton-X-100-containing HBSS, or (**d**) 0.02% saponin-containing HBSS and examined using a fluorescence microscope. (**e**) Quantification of DAPI staining in the fluorescence images (above). A significant difference was obtained via a parametric test but not by using a non-parametric test.

**Figure 3 ijms-22-01040-f003:**
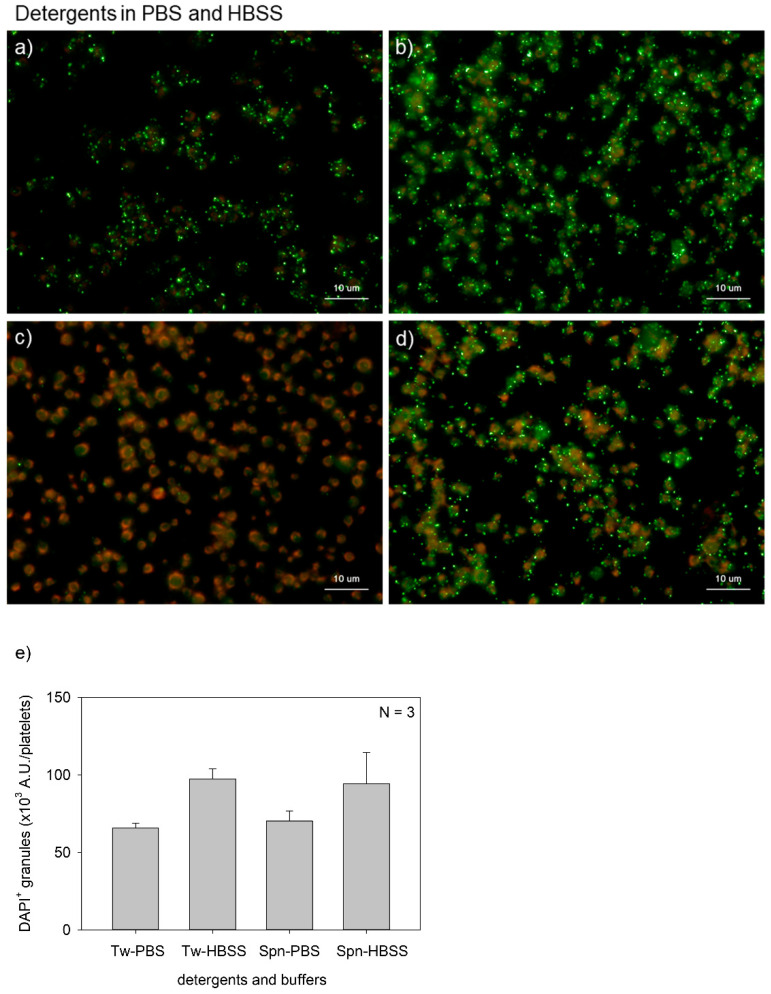
Effects of buffer types on polyphosphate detection in platelets. Platelets were suspended in phosphate-buffered saline (PBS) and incubated for 60 min at room temperature (22–25 °C). Platelets were then immobilized on glass slides using a Cytospin, fixed in 10% neutral-buffered formalin for 4 h, and then treated with 4′,6-diamidino-2-phenylindole (green) and phalloidin (red) in (**a**) 0.1% Tween-20-containing PBS, (**b**) 0.1% Tween-20-containing Hanks balanced salt solution (HBSS), (**c**) 0.02% saponin-containing PBS, or (**d**) 0.02% saponin-containing HBSS and examined using a fluorescence microscope. (**e**) Quantification of DAPI staining in the fluorescence images (above). No significant differences were observed.

**Figure 4 ijms-22-01040-f004:**
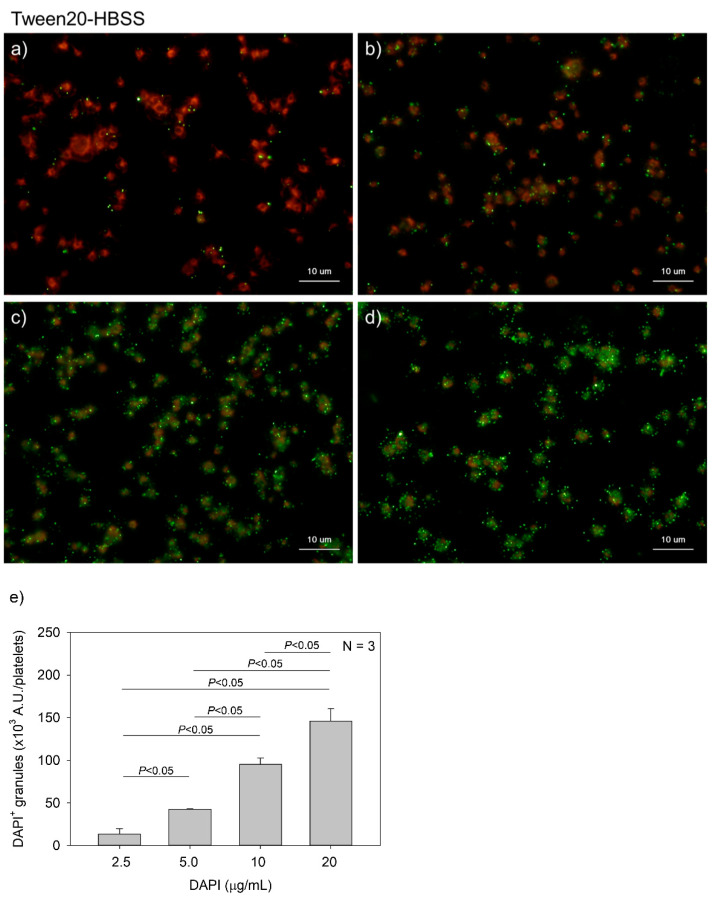
Effects of 4′,6-diamidino-2-phenylindole (DAPI) concentrations on polyphosphate detection in platelets in 0.1% Tween-20-containing Hanks balanced salt solution (Tw-HBSS). Platelets immobilized on glass slides using a Cytospin were fixed in 10% neutral-buffered formalin for 4 h and then treated with DAPI at (**a**) 2.5, (**b**) 5.0, (**c**) 10, or (**d**) 20 μg/mL (green) and phalloidin (red) in Tw-HBSS and examined using a fluorescence microscope. (**e**) Quantitative analysis of the fluorescence images. Significant differences were obtained by a parametric test but not by a non-parametric test.

**Figure 5 ijms-22-01040-f005:**
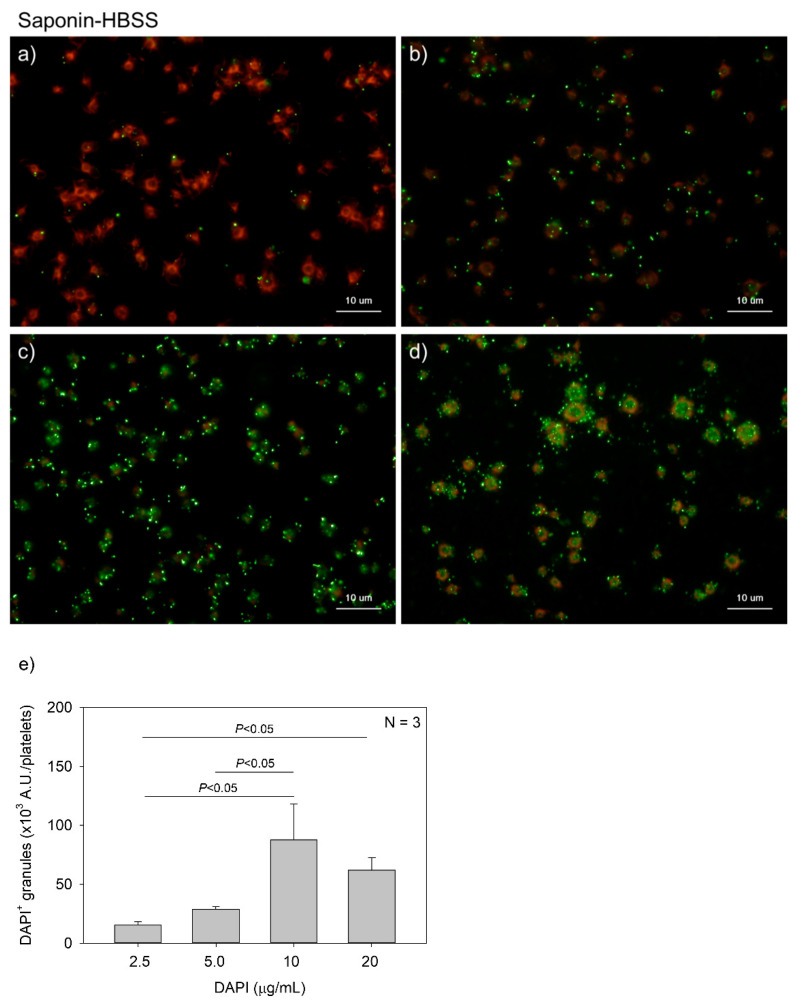
Effects of 4′,6-diamidino-2-phenylindole (DAPI) concentrations on polyphosphate detection in platelets in 0.02% saponin-containing Hanks balanced salt solution (Spn-HBSS). Platelets immobilized on glass slides using a Cytospin were fixed in 10% neutral-buffered formalin for 4 h and then treated with DAPI at (**a**) 2.5, (**b**) 5.0, (**c**) 10, or (**d**) 20 μg/mL (green) and phalloidin (red) in Spn-HBSS and examined using a fluorescence microscope. (**e**) Quantitative analysis of the fluorescence images. Significant differences were obtained by a parametric test but not by a non-parametric test.

**Figure 6 ijms-22-01040-f006:**
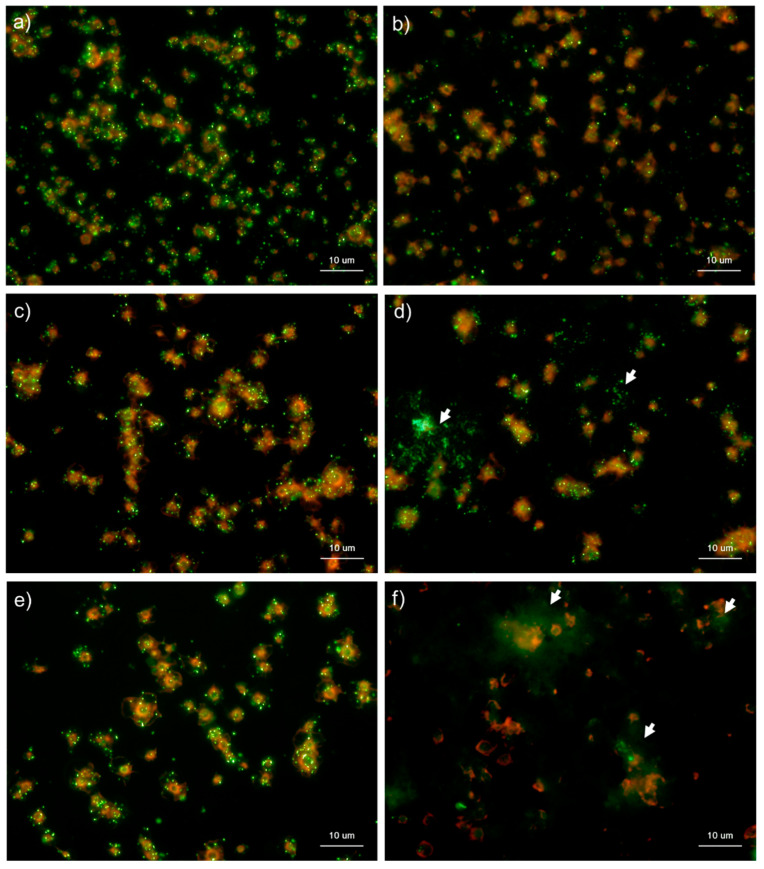
Effects of CaCl_2_ on polyphosphate detection in platelets using 4′,6-diamidino-2-phenylindole (DAPI) in 0.1% Tween-20-containing Hanks balanced salt solution (Tw-HBSS). Platelets suspended in phosphate-buffered saline were activated by 0.1% CaCl_2_ for (**b**) 15 min, (**d**) 30 min, or (**f**) 60 min and then immobilized on glass slides using a Cytospin. Controls (no addition) are shown at (**a**) 15 min, (**c**) 30 min, and (**e**) 60 min. After 4 h of fixation with 10% neutral-buffered formalin, platelets were treated with DAPI (green) and phalloidin (red) in Tw-HBSS and examined using a fluorescence microscope. Arrows indicate possibly diffused polyP. Quantitative analysis could not be performed owing to the aggregation of platelets and diffuse polyP localization.

**Figure 7 ijms-22-01040-f007:**
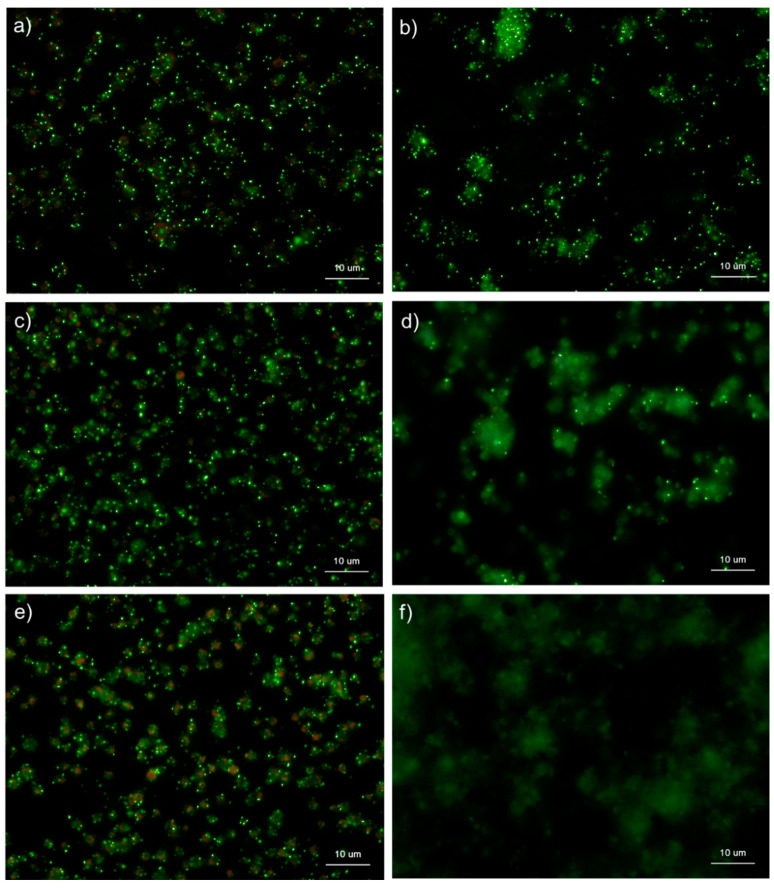
Effects of CaCl_2_ on polyphosphate detection in platelets using 4′,6-diamidino-2-phenylindole (DAPI) in 0.02% saponin-containing Hanks balanced salt solution (Spn-HBSS). Platelets suspended in PBS were activated by 0.1% CaCl_2_ for (**b**) 15 min, (**d**) 30 min, or (**f**) 60 min then immobilized on glass sides using a Cytospin. Controls (no addition) are shown at (**a**) 15 min, (**c**) 30 min, and (**e**) 60 min. After 4 h of fixation with 10% neutral-buffered formalin, platelets were treated with DAPI (green) and phalloidin (red) in Spn-HBSS and examined using a fluorescence microscope. Quantitative analysis could not be performed owing to the aggregation of platelets and diffuse polyP localization.

**Figure 8 ijms-22-01040-f008:**
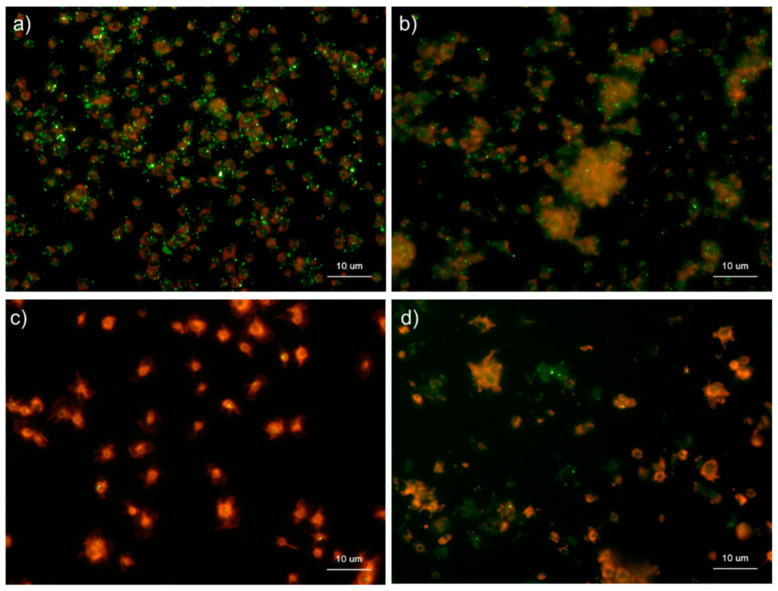
Effects of fixative types on polyphosphate detection in platelets. Platelets suspended in phosphate-buffered saline were activated by 0.1% CaCl_2_ for 60 min (**b**,**d**), immobilized on glass sides using a Cytospin, and fixed with (**a**) 10% neutral-buffered formalin or (**b**) ThromboFix, which was developed and optimized for platelet fixation for the examination of surface markers by flow cytometric analysis. Controls (no addition) are shown in (**a**) 10% neutral-buffered formalin and (**c**) ThromboFix. Platelets were treated with 4′,6-diamidino-2-phenylindole (green) and phalloidin (red) in detergent-free Hanks balanced salt solution and examined using a fluorescence microscope. Quantitative analysis could not be performed owing to the aggregation of platelets and diffuse polyP localization.

**Figure 9 ijms-22-01040-f009:**
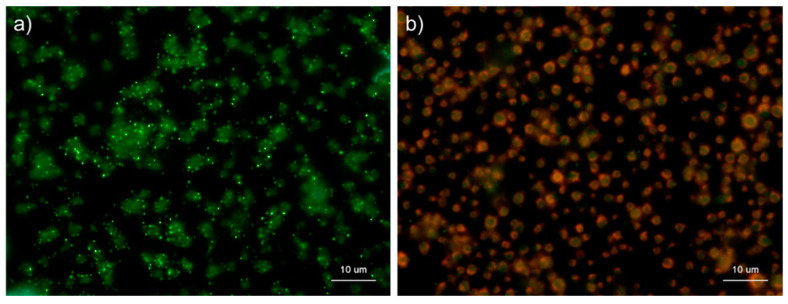
Effect of alkaline phosphatase (ALP) on polyphosphate detection in platelets. Platelets were immobilized on glass sides using a Cytospin. After 4 h of fixation with 10% neutral-buffered formalin, platelets were treated with (**b**) 0.04 U/μL ALP in Mg^2+^-containing alkaline buffer for 24 h, then with 4′,6-diamidino-2-phenylindole (green) and phalloidin (red) in 0.1% Tween-20-containing Hanks balanced salt solution and examined using a fluorescence microscope. The control (no ALP treatment) is shown in (**a**). (**c**) Quantitative analysis of the fluorescence images. Significant differences were observed by both a parametric test and a non-parametric test.

**Figure 10 ijms-22-01040-f010:**
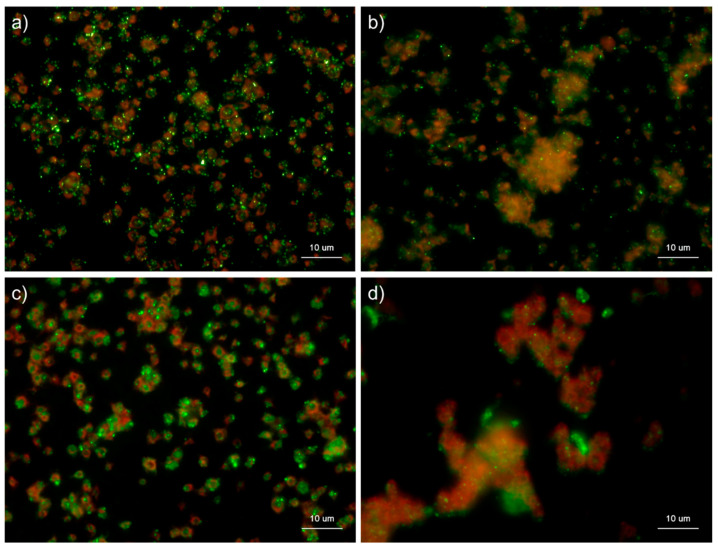
Effects of CaCl_2_ on polyphosphate and serotonin distribution in platelets. Platelets suspended in phosphate-buffered saline (PBS) were activated by 0.1% CaCl_2_ for 60 min (**b**,**d**) then immobilized on glass sides using a Cytospin. After 4 h of fixation with 10% neutral-buffered formalin, platelets were subjected to (**b**) 4′,6-diamidino-2-phenylindole (green) or phalloidin (red) in 0.1% Tween-20-containing Hanks balanced salt solution or (**d**) immunocytochemical examination for serotonin using BlockAce-containing 0.1% Tween-20-containing PBS. The controls (no addition) are shown in (**a**) polyP and (**c**) serotonin. Quantitative analysis could not be performed owing to the aggregation of platelets and diffuse polyP localization.

**Figure 11 ijms-22-01040-f011:**
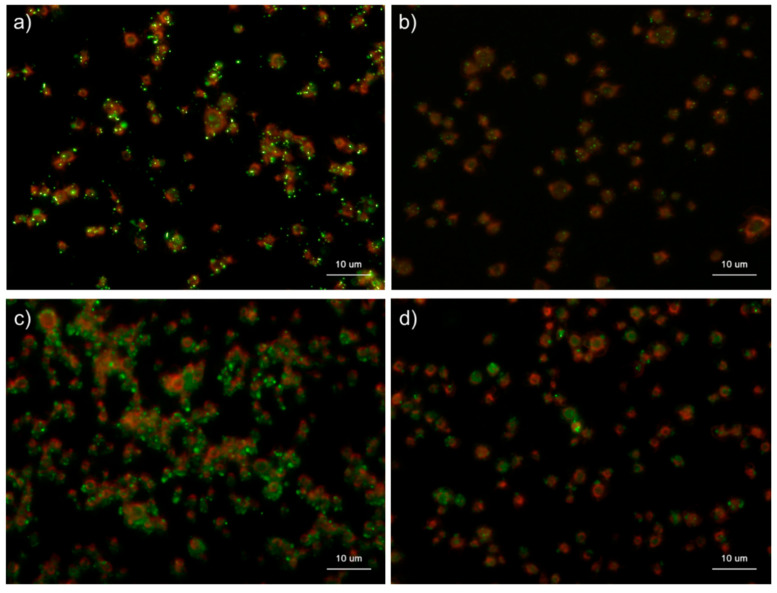
Effects of ADP on the distribution of polyphosphate and serotonin in platelets. Platelets suspended in phosphate-buffered saline (PBS) were activated by 5 mM of ADP for 60 min then immobilized on glass slides using a Cytospin. After 4 h of fixation with 10% neutral-buffered formalin, platelets were subjected to (**b**) 4′,6-diamidino-2-phenylindole (green) or phalloidin (red) in 0.1% Tween-20-containing Hanks balanced salt solution or (**d**) immunocytochemical examination for serotonin using BlockAce-containing 0.1% Tween-20-containing PBS. The controls (no addition) are shown in (**a**) polyP and (**c**) serotonin. Quantitative analysis could not be performed owing to the aggregation of platelets and diffuse polyP localization.

## Data Availability

The datasets used and/or analyzed during the current study are available from the corresponding author on reasonable request.

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
