# Peer review of "Fluorescent Cytochemical Detection of Polyphosphates Associated with Human Platelets"

_ijms, 2021, doi:10.3390/ijms22031040_

Round 1

Reviewer 1 Report

The study by Sato et al investigates the amount of Polyphosphate in fixed human platelets in suspension that were immobilized by a cytospin. The study is well-performed and the images support, that polyphosphate can be detecetd semiquantitavely in dense granules of platelets. I belive the data.

However, the concept is not very well-known in medicine, only dentists and dental surgeons seem to use platelet-rich plasma for a platelet-rich-fibrin therapy. Also veterenitarians use ist for horses.

I just do  not know, if this is a generally accepted therapy or if it belongs to the stories of phantasia-country. The editors should ask a reviewer from the dental field as well. In this case, a journal for dental medicine would probably suit better for this paper.

Author Response

The study by Sato et al investigates the amount of Polyphosphate in fixed human platelets in suspension that were immobilized by a cytospin. The study is well-performed and the images support, that polyphosphate can be detecetd semiquantitavely in dense granules of platelets. I belive the data.

However, the concept is not very well-known in medicine, only dentists and dental surgeons seem to use platelet-rich plasma for a platelet-rich-fibrin therapy. Also veterenitarians use ist for horses.

I just do not know, if this is a generally accepted therapy or if it belongs to the stories of phantasia-country. The editors should ask a reviewer from the dental field as well. In this case, a journal for dental medicine would probably suit better for this paper.

Response: Thank you for your helpful comments. We do not exactly understand what you mean by “the stories of fantasia-country.” PRP therapy is applied not only in the dental field but also in other medical fields, such as orthopedic surgery, plastic surgery, dermatology, and esthetic surgery. As mentioned in the text, however, polyP functions in tissue regeneration when applied as a form of PRP is still poorly understood. Thus, we aimed to prove the possible involvement of polyP in tissue regeneration in a step-by-step manner. We do not think that this information should be shared only in the dental field.

At present, it is generally thought that polyP facilitates coagulation and maturation of fibrin fibers, effects that are involved in retention and controlled release of growth factors derived from platelets. Thus, it is expected that the quantity and quality of polyP contained in individual PRP preparations may influence the quality of individual PRP preparations and probably their clinical outcomes

Reviewer 2 Report

The authors report and describe, in experiments with human platelets, a fluorescent cytochemical method to visualize poly-P in platelets. Using a treatment of platelets with 4', 6- diamidino-2-phenylindole (DAPI), the authors claim that the DAPI-targets are poly-phosphates and that their method can be used for the semiquantitative determination of the amounts and distribution of poly-P in platelets.

Polyphosphate(s) recently received an extraordinary attention in the fields of hemostasis and platelets, since it is a procoagulant inorganic polymer of linear linked orthophosphates. Multiple recent and very recent investigations established the importance of platelet polyphosphate in blood coagulation; however, the mechanistic details of polyphosphate homeostasis in mammalian species remain largely undefined. Therefore, a semiquantitative determination of amounts and distribution of poly-P would be very valuable. Unfortunately, the results of this paper describe only a cytochemcal detection method without quantitation and appropriate chemical/biological controls.

  • No quantitative data of the various stains are reported.
  • What is the chemical evidence that DAPI sees poly-phosphates (and only that) in their experiments? Some chemical validation (more than the use of ALP) is required.
  • The authors report some kind of correlation of DAPI staining and dense granules, but no further controls were provided (time dependent depletion of dense granules, platelets deficient of dense granules and others). Were the platelets really stimulated with 5 mM ADP? 10 µM should do it.
  • The authors should also consult the very recent (2019/2018) literature on the topic platelet polyphosphates.   

Author Response

The authors report and describe, in experiments with human platelets, a fluorescent cytochemical method to visualize poly-P in platelets. Using a treatment of platelets with 4', 6- diamidino-2-phenylindole (DAPI), the authors claim that the DAPI-targets are poly-phosphates and that their method can be used for the semiquantitative determination of the amounts and distribution of poly-P in platelets.

1) Polyphosphate(s) recently received an extraordinary attention in the fields of hemostasis and platelets, since it is a procoagulant inorganic polymer of linear linked orthophosphates. Multiple recent and very recent investigations established the importance of platelet polyphosphate in blood coagulation; however, the mechanistic details of polyphosphate homeostasis in mammalian species remain largely undefined. Therefore, a semiquantitative determination of amounts and distribution of poly-P would be very valuable. Unfortunately, the results of this paper describe only a cytochemcal detection method without quantitation and appropriate chemical/biological controls.

No quantitative data of the various stains are reported.

Response: Thank you for this comment. We tried to quantify the present cytochemical data. However, we found that it was difficult to normalize DAPI-positive substances by other markers such as F-actin, because the levels of polymerized actin depend on the activation status of platelets. In addition, it was difficult to accurately quantify the data using conventional image analysis when platelets were aggregated upon activation. 

It should be noted that normalization with DNA content, which was performed in several published articles, such as ref. #16 [Gomes et al., 2013], cannot be applied to anuclear cells, such as platelets. Data normalization was also performed in extracted polyP in the study [Verhoef et al., Blood 2017]. However, details of the normalization protocol were not provided.

It is also known that platelets contain RNA [Rowley et al., Curr Opin Hematol 2012; Sol et al., Cancer Metastasis Rev 2017], which may bind to polyP and thereby interfere with accurate polyP quantification, depending on its levels, although it could be used as a factor for normalization. We examined our specimens but did not find apparent RNA-dependent fluorescence signals, unlike DNA-dependent signals which were observed at a excitation/emission wavelength of 360/460 nm in platelets. It is possible that unlike DNA, RNA contents may vary with activation status of platelets [Wurzinger, Adv Anat Embryol Cell Biol 1990; Fox, Thromb Haemost 1993], thus even though RNA can be detected by DAPI, it may be difficult to apply RNA for normalization of visualized polyP contents.

Instead, we are now developing and optimizing the methods to quantify polyP contents using fluorometric and flow cytometric analyses. Because we have not yet completed it, we believe that such preliminary data should not be disclosed at present. However, Reviewer #3 also required us to present the quantitative data. Thus, we have added the preliminary data as a supplemental material only for your evaluation. It should be noted that the measured fluorescence intensity is currently normalized by platelet counts in our current project.

2) What is the chemical evidence that DAPI sees poly-phosphates (and only that) in their experiments? Some chemical validation (more than the use of ALP) is required.

Response: This is a good point. Several probes have been developed for polyP detection. However, judging from the literature, most of them are not commercially available or have lower specificity to polyP. Thus, to our knowledge, ALP digestion is the only convenient way to support the specificity of DAPI-binding.

3) The authors report some kind of correlation of DAPI staining and dense granules, but no further controls were provided (time dependent depletion of dense granules, platelets deficient of dense granules and others). Were the platelets really stimulated with 5 mM ADP? 10 µM should do it.

Response: Thank you for this comment. We thought of a similar possibility during the experimental design. To prove the depletion of dense granules, TEM examination should be performed. However, we did not have access to TEM for use in this study.

In addition, platelets that are fully activated to deplete dense granules are usually aggregated and not suitable for subsequent image analysis. Thus, we did not pursue this option.

We fixed ADP doses that were used at 5 mM ,based on data from our previous studies [Tsujino et al., Dent J 7(2):61; 2019], and other literature. We have confirmed that ADP at 10 µM is capable of aggregating platelets but not enough to significantly release growth factors and polyP.

4) The authors should also consult the very recent (2019/2018) literature on the topic platelet polyphosphates.

Response: We cited three papers published in 2013–2020 (ref. # 14–16) at the beginning of the Discussion section. To our knowledge, there has been no substantial progress in methodology for polyP visualization using DAPI or other probes.

Reviewer 3 Report

In this manuscript the authors established a cytochemical method to visualise PolyP in platelets using DAPI staining. They optimised the staining procedure for fixation time; buffer, fixative and detergent type; and DAPI concentration. In addition they showed PolyP mobilisation in calcium- or ADP-treated platelets and PolyP and serotonin distribution in calcium-treated platelets. The specificity of the DAPI staining for PolyP was confirmed by addition of alkaline phosphatase to the platelets, cleaving PolyP.

The manuscript gives an excellent overview on the methodology to visualise PolyP inside platelets. The authors state that their method is useful for the semi quantitative determination of the amounts and distribution of PolyP in platelets. However, to this reviewers’ opinion they do not provide substantial evidence for this statement. No quantification (eg of fluorescence amount or intensity) is performed, nor are comparisons made for PolyP amount between for example healthy individuals, specific patient groups (eg storage pool deficiency or other) or in vitro treatment. In light of the clinical relevance stated in the manuscript, such confirmation beyond validation of the method would be considered essential.

Following the comment on semi quantitative assessment of PolyP content in platelets, information would be required on the origin of samples used. When comparing images in Figures which have been taken from samples processed similarly (ie control samples, Fig 1d, 2b, 3b, 4c etc, or calcium treated samples in Fig 6f, 10b) clear visual differences can be seen in PolyP amount and distribution. Hence, the question arises what causes these differences. Is this donor specific; is the staining pattern and intensity consistent when the same donor is used? In addition, phalloidin is used to visualise platelet actin: how come there are large differences in phalloidin staining and what does this say about the quality of DAPI stain? For example, Fig 4 and 5 show great differences in phalloidin staining while the only variable in a-d is the DAPI concentration?

The need for slight permeabilisation of the platelet membrane to allow intercellular PolyP staining is highlighted by the absence of detergent in Fig 2a. Is the diffuse DAPI staining minimally present here coming from extracellularly bound PolyP? And how can this extracellular component be excluded when visualising intracellular PolyP?

Methodically the authors have optimised many of the steps for PolyP visualisation in platelets using DAPI. The one variable that seems to be missing is the actual staining time. Does this affect the staining quality?

The authors indicate other groups have used DAPI staining in order to visualize PolyP intracellularly, but do not indicate how their findings relate to this published data. The cited manuscript: “Gomes F.M.; Ramos, I.B.; Wendt, C.; Girard-Dias, W.; De Souza, W.; Machado, E.A.; Miranda, K. New insights into the in situ microscopic visualization and quantification of inorganic polyphosphate stores by 4',6-diamidino-2-phenylindole (DAPI)-staining. European journal of histochemistry : EJH. Volume 57; 2013. p e34.” shows intracellular PolyP in different celtypes and emphasises the need for different methodology.

Ref “Verhoef J.J.; Barendrecht, A.D.; Nickel, K.F.; Dijkxhoorn, K.; Kenne, E.; Labberton, L.; McCarty, O.J.; Schiffelers, R.; Heijnen, H.F.; Hendrickx, A.P.; Schellekens, H.; Fens, M.H.; de Maat, S.; Renné, T.; Maas, C. Polyphosphate nanoparticles on the platelet surface trigger contact system activation. Blood. 331 2017, 129, 1707-1717.” shows PolyP visualisation in platelets.

Also, both introduction and discussion should include and not overlook alternative PolyP staining methods (and possibly their shortcomings) to indicate the value of this work.

Minor:

Platelets promote coagulation and ensuing fibrin formation by exposing phosphatidylserine on their surface, this is overlooked in the introduction.  

Figure 10 and 11 legend: line 183 and 196, respectively: (b) should be (c).

Materials and Methods 5.2 seems to only state the eventual optimal method. However, for formalin treatment 30 minutes is noted instead of 4h.

Author Response

In this manuscript the authors established a cytochemical method to visualise PolyP in platelets using DAPI staining. They optimised the staining procedure for fixation time; buffer, fixative and detergent type; and DAPI concentration. In addition they showed PolyP mobilisation in calcium- or ADP-treated platelets and PolyP and serotonin distribution in calcium-treated platelets. The specificity of the DAPI staining for PolyP was confirmed by addition of alkaline phosphatase to the platelets, cleaving PolyP.

1) The manuscript gives an excellent overview on the methodology to visualise PolyP inside platelets. The authors state that their method is useful for the semi quantitative determination of the amounts and distribution of PolyP in platelets. However, to this reviewers’ opinion they do not provide substantial evidence for this statement. No quantification (eg of fluorescence amount or intensity) is performed, nor are comparisons made for PolyP amount between for example healthy individuals, specific patient groups (eg storage pool deficiency or other) or in vitro treatment. In light of the clinical relevance stated in the manuscript, such confirmation beyond validation of the method would be considered essential.

Response: It was difficult to find appropriate ways to normalize the quantification of DAPI-positive substances. Thus, we did not show the quantitative data in the original version. Instead, for your evaluation, we present preliminary data from our currently ongoing project using a fluorometer as a supplemental material. Because we have not yet fully optimized the procedure and because the platelet conditions (i.e., adhered vs. suspended) are different, it is difficult to directly compare these data with the cytochemical data. However, we believe that activation with CaCl2 significantly reduces the intracellular (and probably also membrane-bound) contents of polyP in a dose-dependent manner in the fluorometric data.

2) Following the comment on semi quantitative assessment of PolyP content in platelets, information would be required on the origin of samples used. When comparing images in Figures which have been taken from samples processed similarly (ie control samples, Fig 1d, 2b, 3b, 4c etc, or calcium treated samples in Fig 6f, 10b) clear visual differences can be seen in PolyP amount and distribution. Hence, the question arises what causes these differences. Is this donor specific; is the staining pattern and intensity consistent when the same donor is used? In addition, phalloidin is used to visualise platelet actin: how come there are large differences in phalloidin staining and what does this say about the quality of DAPI stain? For example, Fig 4 and 5 show great differences in phalloidin staining while the only variable in a-d is the DAPI concentration?

Response: Thank you for this excellent question. This is the main point of optimizing the procedure. We used the same concentrations of the reagents and blood samples from the same donors, at least in the same figures. The processing was performed similarly throughout this project.

Here is our answer to your question. After incubation with CaCl2, platelets are activated and increase their ability to adhere to the surrounding environment. Thus, activated platelets adhere easily and quickly onto the glass slide during centrifugation (within 2 min) and form filopodium and lamellipodium to flatten, but enlarges their size with well-developed F-actin. However, when platelets aggregate, actin polymerization can be suppressed and the phalloidin-dependent fluorescence signal is substantially reduced.

Both Saponin and Tween-20 permeabilize the plasma membrane. However, it is generally thought that their ability to do so is different: Saponin is milder than Tween-20, and so saponin extracts and deprives relatively lower amounts of membrane components, such as proteins or phospholipids, than Tween-20. Thus, we believe that DAPI-positive substances could be retained at higher levels in saponin-treated platelets.

Although these detergents may influence the membrane permeability of phalloidin, it is possible that the phalloidin-dependent fluorescence signal is reduced when DAPI-positive substances are retained in the cytoplasm at higher levels.

3) The need for slight permeabilisation of the platelet membrane to allow intercellular PolyP staining is highlighted by the absence of detergent in Fig 2a. Is the diffuse DAPI staining minimally present here coming from extracellularly bound PolyP? And how can this extracellular component be excluded when visualising intracellular PolyP?

Response: It is known that low amounts of methanol are added to preserve formalin in commercially available formalin solutions. Thus, we suggest the possibility that the membrane is slightly permeabilized during fixation. However, we should also consider the possibility that platelets could be slightly activated during centrifugation to release some polyP into the extracellular spaces. Some polyP may be bound to the outer surface of the membrane to remain with platelets and not diffuse away.

As shown in Figure 8c, it is better to use ThromboFix for platelet fixation to minimize intracellular polyP. In other words, ThromboFix can exclude possible contamination of intracellular polyP in the control platelets. Thus, we believe that all these images should be present in combination. To our knowledge, there is no way to specifically and efficiently dissect intracellular polyP from extracellular polyP in cytochemical visualization.

4) Methodically the authors have optimised many of the steps for PolyP visualisation in platelets using DAPI. The one variable that seems to be missing is the actual staining time. Does this affect the staining quality?

Response: Judging from our experience, it is not necessary to treat cells with dyes for longer time periods in polyP and F-actin staining. These polymerized targets can theoretically be stained quickly and are hardly overstained by prolonged staining time (~1 h). However, membrane permeabilization and delivery of dyes take several minutes. Thus, we chose 30 min to maximize the specific staining through appropriate membrane permeabilization in one step.

5) The authors indicate other groups have used DAPI staining in order to visualize PolyP intracellularly, but do not indicate how their findings relate to this published data. The cited manuscript: “Gomes F.M.; Ramos, I.B.; Wendt, C.; Girard-Dias, W.; De Souza, W.; Machado, E.A.; Miranda, K. New insights into the in situ microscopic visualization and quantification of inorganic polyphosphate stores by 4',6-diamidino-2-phenylindole (DAPI)-staining. European journal of histochemistry : EJH. Volume 57; 2013. p e34.” shows intracellular PolyP in different celtypes and emphasises the need for different methodology.

Ref “Verhoef J.J.; Barendrecht, A.D.; Nickel, K.F.; Dijkxhoorn, K.; Kenne, E.; Labberton, L.; McCarty, O.J.; Schiffelers, R.; Heijnen, H.F.; Hendrickx, A.P.; Schellekens, H.; Fens, M.H.; de Maat, S.; Renné, T.; Maas, C. Polyphosphate nanoparticles on the platelet surface trigger contact system activation. Blood. 331 2017, 129, 1707-1717.” shows PolyP visualisation in platelets.

Also, both introduction and discussion should include and not overlook alternative PolyP staining methods (and possibly their shortcomings) to indicate the value of this work.

Response: Thank you for this comment. We have added some notes referring to the above articles in the last paragraph of the Introduction section to further explain the historical background and highlight the current state of methodological development and have included additional references. In addition, to expand this note, we have added a paragraph on the technical limitations, that is, normalization, of our optimized method in the Discussion section and pointed out the advantage of this method in the section on clinical relevance.

Minor:

1) Platelets promote coagulation and ensuing fibrin formation by exposing phosphatidylserine on their surface, this is overlooked in the introduction. 

Response: Thank you for the helpful advice. We have added this phrase to line 45.

2) Figure 10 and 11 legend: line 183 and 196, respectively: (b) should be (c).

Response: Thank you for pointing this out. We have corrected these sections.

3) Materials and Methods 5.2 seems to only state the eventual optimal method. However, for formalin treatment 30 minutes is noted instead of 4h.

Response: Thank you for pointing this out. We have corrected this section.

Round 2

Reviewer 2 Report

The authors addressed most of the comments/problems raised by this and other referees. However, no new experiments, controls and data were presented.

Clearly, no quantitative data were provided. Any statement that the data are quantititative or even semi-quantitative should be removed.

Also, the nature of the compound(s) responsible for the cytochemical signals is unclear. It could be due to polyphosphates, but other possibilities exist. No further validation was provided, no alternative methods.This should be indicated.

The cell biology of the staining was also only superficially addressed. The granule nature could be studied in more detail. 

Overall, the paper is entirely descriptive with no molecular insights at present.  

Author Response

- The authors addressed most of the comments/problems raised by this and other referees. However, no new experiments, controls and data were presented. Clearly, no quantitative data were provided. Any statement that the data are quantititative or even semi-quantitative should be removed.

Response: As you may know, we were given only 10 days for this major revision, including English editing. Thus, we had to carefully consider what could be done in this limited period. We performed image analysis to quantify the brightness of DAPI staining in polyP (and possibly also some similar compounds). The data were normalized to the number of platelets, which were visualized via actin staining. Because the pure platelet suspension does not include nucleated cells, such as leukocytes, even though DAPI binding is not as specific as that of antibodies, we think it is still necessary to consider possible interference by DNA.

As for RNA and ATP (also ADP) in platelets, we do not think that polyP visualization is significantly influenced by these molecules because of their instability and the relatively less quantity, as assessed via phosphate counts.

Thus, we think that these data are quantitative enough to use “semi-quantitative”’ in the text. However, to address your specific concern, we have deleted this word from the abstract.

- Also, the nature of the compound(s) responsible for the cytochemical signals is unclear. It could be due to polyphosphates, but other possibilities exist. No further validation was provided, no alternative methods. This should be indicated.

The cell biology of the staining was also only superficially addressed. The granule nature could be studied in more detail.

Response: Unfortunately, we are unsure on how to appropriately address these comments. To the best of our knowledge, there are no highly specific, reproducible, and commercially available probes for polyP detection. We have described such limitations in the introduction section (lines 60–67).

Our research expertise is in the use of platelet concentrates for regenerative therapy; however, we are less familiar with detailed techniques used to research polyP specifically. Thus, we would appreciate it if you kindly provided suggestions and/or references to studies that you believe would appropriately improve our work. Even if this article is eventually rejected, we are very interested in incorporating expert advice that can improve the quality and impact of our studies.

- Overall, the paper is entirely descriptive with no molecular insights at present.

Response: This is a methodology paper in terms of the most frequently used polyP visualization methods. The goal of our research project is to develop methods to ensure the quality of platelet concentrates for regenerative therapy at the point-of-care. Thus, our study is based on previous DAPI-polyP studies that you may not accept. In addition, the improvements we made regarding the presentation of quantified results per your previous comment expand this work and do not limit it to a merely descriptive study. Finally, we believe that in the context of methodology development, detailed molecular studies are not necessary at this time but can be addressed in follow-up work.

Reviewer 3 Report

The authors have primarily responded to the comments and suggestions textually, with no (essential) additional experiments. One of the main concerns about the quantification of the staining (as it is key for controlling and assessing the quality of the results) remains. The authors tried to tackle this by a fluorometric assay. However, using an other method does not provide the required information on the topic of quantification of the DAPI staining as intented in the manuscript. Hence, no comments in the manuscript about the (poossible) quantification can be considered valid.  

Furthermore, no validation of the results by means of patients or appropriate inhibitors was performed (granule defect of release).

The paper gives an excellent method for polyp staining, but lacks novelty in the research area. Additional experiments are crucial to provide validation and novelty.

Author Response

- The authors have primarily responded to the comments and suggestions textually, with no (essential) additional experiments. One of the main concerns about the quantification of the staining (as it is key for controlling and assessing the quality of the results) remains. The authors tried to tackle this by a fluorometric assay. However, using an other method does not provide the required information on the topic of quantification of the DAPI staining as intented in the manuscript. Hence, no comments in the manuscript about the (poossible) quantification can be considered valid.

Response: As described in the introduction section, even though DAPI is not specific to polyP-like antibodies, there are no other ways to conveniently detect or visualize polyP using the commercially available probes. Thus, we can reduce the possible interfering substances and possible non-specific binding by optimizing the experimental conditions. In this study, we successfully optimized the protocol to visualize polyP within such conditions.

In this second revision, we quantified the brightness of the DAPI signal and normalized the data with platelet counts. We hope that you will recognize that the data is now presented quantitatively. We also appreciate your advice as it is helpful for our project.

- Furthermore, no validation of the results by means of patients or appropriate inhibitors was performed (granule defect of release).

Response: In our experimental plan, which was approved by the ethics committee, we did not collect blood samples from patients. In fact, we are not authorized to select patients who lack functioning platelets, as only someone designated as a hospital faculty member can do that.

As addressed above, our goal in this study was to develop an effective and rapid methodology for staining polyP in platelets. We are interested and willing to further investigate the molecular mechanisms of DAPI-stained polyP in platelet granules and the suggestions that you have made are valid in that context. However, we believe that the results presented in this manuscript sufficiently demonstrate a quality protocol for rapid, effective, and high-quality staining of polyP with DAPI.

- The paper gives an excellent method for polyp staining, but lacks novelty in the research area. Additional experiments are crucial to provide validation and novelty.

Response: We performed image analysis to provide quantitative data. That is all we were able to complete within the 10 days given to us for revision.

Round 3

Reviewer 3 Report

I value that the authors have added the quantification on the fluorescence intensity of the images.  

Additional experiments were not possible with patient material, but unexplored for pharmacological in vitro interventions.

I appreciate that with 10 days there is a limitation on the amount of work that can be performed.